# Augmented CPT1A Expression Is Associated with Proliferation and Colony Formation during Barrett’s Tumorigenesis

**DOI:** 10.3390/ijms231911745

**Published:** 2022-10-04

**Authors:** Joshua N. Bernard, Vikram Chinnaiyan, Thomas Andl, Gregoire F. Le Bras, M. Nasar Qureshi, Deborah A. Altomare, Claudia D. Andl

**Affiliations:** 1Burnett School of Biomedical Sciences, University of Central Florida, Orlando, FL 32816, USA; 2QDx Pathology Services, Edison, NJ 08837, USA

**Keywords:** obesity, fatty acid oxidation, high fat diet, gastroesophageal reflux diseases (GERD), inflammation

## Abstract

Obesity is a known risk factor for the development of gastroesophageal reflux disease (GERD), Barrett’s Esophagus (BE) and the progression to esophageal adenocarcinoma. The mechanisms by which obesity contributes to GERD, BE and its progression are currently not well understood. Recently, changes in lipid metabolism especially in the context of a high fat diet have been linked to GERD and BE leading us to explore whether fatty acid oxidation plays a role in the disease progression from GERD to esophageal adenocarcinoma. To that end, we analyzed the expression of the rate-limiting enzyme, carnitine palmytoyltransferase 1A (CPT1A), in human tissues and cell lines representing different stages in the sequence from normal squamous esophagus to cancer. We determined uptake of palmitic acid, the most abundant fatty acid in human serum, with fluorescent dye-labeled lipids as well as functional consequences of stimulation with palmitic acid relevant to Barrett’s tumorigenesis, e.g., proliferation, characteristics of stemness and IL8 mediated inflammatory signaling. We further employed different mouse models including a genetic model of Barrett’s esophagus based on IL1β overexpression in the presence and absence of a high fat diet and deoxycholic acid to physiologically mimic gastrointestinal reflux in the mice. Together, our data demonstrate that CPT1A is upregulated in Barrett’s tumorigenesis and that experimental palmitic acid is delivered to mitochondria and associated with increased cell proliferation and stem cell marker expression.

## 1. Introduction

Obesity has been shown to be an independent risk factor for gastro-esophageal reflux disease and esophageal adenocarcinoma [1,2]. Interestingly, the number of children diagnosed with GERD is increasing [3] and has been described to be associated with the intake of higher calories and larger amounts of fat [4]. More so, in that cohort, patients with erosive GERD had a higher intake of energy, protein, and total fat and a lower intake of polyunsaturated fats than those with non-erosive GERD [4]. Cohort studies demonstrate that symptoms of GERD occur in 20–50% of US adults [5,6]. Multiple large case–control studies demonstrated a positive association between reflux symptoms, Barrett’s esophagus and the risk of esophageal adenocarcinoma (EAC), especially in the context of prolonged and severe symptoms [6,7]. Interestingly, increasing rates of EAC have paralleled the rise in obesity [8,9]. This increased risk of obese individuals can be attributed to several factors including the high prevalence of GERD, and a linear association between central adiposity and Barrett’s esophagus [9,10], a premalignant condition whereby the normal squamous epithelium is replaced by columnar lined epithelium with intestinal metaplasia [6]. While adiposity physically exerts abdominal pressure on the gastro-esophageal sphincter leading to gastroesophageal reflux (GERD) [11], obesity has also been associated with chronic low-grade inflammation through elevated levels of circulating saturated fatty acids, which trigger inflammatory responses [9]. Interestingly, the cell signaling of high leptin and low adiponectin levels can alter the capacity of cell proliferation (reviewed by Schlottman et al. [9]). These findings are further supported by recent animal studies showing that non-transgenic mice placed on a high fat diet develop esophageal inflammation and metaplasia at 9 months [12]. Very-long-chain ceramides and reduced lysophospholipids characterized a tissue lipid signature for this model of Barret’s esophagus [12]. A high fat diet also accelerated the development of esophageal dysplasia and tumors, independent of body weight in a mouse model with targeted overexpression of the pro-inflammatory cytokine IL1β [13]. Additionally, an increased number of neutrophils to natural killer cells and a shift in the gut microbiome were observed during the progression to cancer [13]. Mechanistically, it appears that increased cytokines, including IL8, are at the center of the high fat diet-induced effects such as inflammation and stem cell expansion, independent of obesity [13].

It is widely accepted that inflammation can increase the risk for cancer, and this link has been studied in connection with Barrett’s tumorigenesis as well [14,15]. Oxidative stress, the activation of NF-kB and DNA damage have been highlighted as molecular pathways contributing to the progression of Barrett’s esophagus to adenocarcinoma [15,16,17,18,19]. It is now well known that obesity is a state of chronic low-grade, systemic inflammation, also termed “meta-inflammation”, predominantly caused by the secretion of a variety of pro-inflammatory mediators by the fat tissue: it is believed these cytokines contribute to the increased risk for many cancers associated with obesity [20,21].

Carnitine palmitoyltransferase 1A (CPT1A), the rate-limiting enzyme of mitochondrial fatty acid oxidation (FAO), has emerged as a key molecule in different cancers [22,23]. CPT1A mediates the transport of long-chain fatty acid from the cytoplasm into mitochondria for oxidation, thereby initiating a cyclical series of reactions that result in the shortening of fatty acids [20,21,22]. These reactions generate NADH and FADH2 in each round, which upon entering the electron transport chain produce ATP, directly resulting in increased cell survival and cell proliferation, with direct relevance to the metabolism of cancer cells [22,23,24]. CPT1A has been reported to be upregulated in numerous cancers, including breast cancer [25,26,27]. Its function has been described to play a role in lymphangiogenesis [28], intestinal stemness [29,30], and colorectal metastasis [27]. Located on 11q13, a frequently mutated area in esophageal squamous cancer, and nearby Cyclin D1, CCND1, which is overexpressed in esophageal adenocarcinoma [31,32], CPT1A has been identified as a marker for poor survival in esophageal squamous cell carcinoma [33]. 

These observations prompted us to explore the abundance and role of CPT1A and FAO along the progression from normal esophageal squamous to EAC. To determine its contribution to the pathogenesis of the disease, we utilized human cell lines as well as mouse and human tissues. To assess growth and proliferation in the context of increased availability of free fatty acids (FFA), we utilized the same human cell lines along the spectrum of Barrett’s tumorigenesis, e.g., CPA, BAR-T, OE33 and FLO1, and additionally two different mouse models. We identified CPT1A to be the dominant isoform expressed in the esophageal cell lines, and the ability of these cells for uptake of palmitic acid, the most abundant FFA in human serum. Here, we also demonstrate that palmitic acid treatments, used to mimic the excess of FFA in high fat diets and obesity, promotes cell proliferation, inflammatory signaling and the expression of progenitor or stem cell markers.

## 2. Results

### 2.1. Expression of CPT1A Is the Predominant CPT1 Isoform Expressed in the Esophagus and Upregulated in Barrett’s Tumorigenesis

To determine the expression of the three carnitine palmitoyltransferase (CPT1) isoforms in the progression from squamous epithelium to EAC, we queried the public GEO datasets, GSE34619 (Figure 1a) and GDS1321/GSE1420 (Figure 1b). In both data sets, we observed a marked increase of CPT1A, which mediates the transport of long-chain fatty acids from the cytoplasm into mitochondria for oxidation, on an mRNA level between normal tissue and BE (*p* < 0.05), and a trend towards a further increase in EAC cells (Figure 1b). CPT1B, another isoform, was significantly enriched in BE compared to normal tissue in only one dataset (GSE34619, Figure 1a). Analysis of other components of fatty acid metabolism such as CD36, a fatty acid translocase which binds long-chain fatty acids and facilitates their transport into cells, showed no significant increase in the same test sets (Figure 1a,b). CPT1A is upregulated in human BE and EAC tissues compared to the adjacent normal esophageal epithelium, a healthy esophageal control as quantified by signaling intensity and epithelial location using a composite score (Figure 1c) based on immunofluorescence staining (representative images in Figure 1d). To confirm the expression of CPT1A and other CPT1 isoforms, we analyzed a panel of cell lines representing the sequences from normal squamous esophagus to EAC. Additionally, we used hTERT-immortalized esophageal cells as control (STR) and STR tolerant to exposure with pH4/bile salts mimicking GERD [34]. Representing Barrett’s esophagus, we used the non-dysplastic CPA cell line and BAR-T [35]. The EAC cell lines were OE33 and FLO1. CPT1A is the dominant isoform expressed in the panel of experimental cell lines tested, with only CPT1B being elevated in BAR-T cells as evaluated by qRT-PCR (Appendix A). mRNA levels were lower in BE cell lines but increased in the EAC cell lines: OE33 and FLO1 (Figure 2a). We observed higher protein expression of CPT1A in the GERD model, STR B/A, two out of the three BE cell lines (BAR-T and BAR-T10T) and three EAC cell lines compared to normal squamous STR (Figure 2b) by Western Blot. We further demonstrated that the inhibition of CPT1A function using the chemical compound etomoxir results in a compensatory feedback loop augmenting CPT1A protein expression in an effort to restore function (Appendix A). Etomoxir (ethyl 2-[6-(4-chlorophenoxy)hexyl]oxirane-2-carboxylate), has been shown to pharmacologically target FAO and is regarded as a specific inhibitor of CPT1 [36,37].

### 2.2. Esophageal Epithelial Cells Have the Capacity for Palmitic Acid Uptake and Incorporation into Mitochondria

Plasma concentrations of free fatty acids (FFA) are usually found to be elevated in obese individuals [38,39]. Plasma FFA easily translocate into the cytoplasm of cells where they are oxidized to generate energy in the form of ATP or re-esterified for storage as triglycerides. Not surprisingly, raising blood FFA levels results in the accumulation of triglycerides, usually in muscle or hepatic cells [40,41]. The most abundant fatty acid is oleic acid followed by palmitic acid [42]. Saturated fatty acids and monounsaturated fatty acids elicit hypercholesterolemia effects. Palmitic acid (PA) is the most common saturated fatty acid accounting for 20–30% of total fatty acids in the human body and can be provided in the diet or synthesized endogenously via de novo lipogenesis [42]. It participates as a membrane phospholipid in providing membrane structure and function but also makes up adipose triacylglycerols [42].

To evaluate the effects of FFAs in the context of a high fat diet, we treated esophageal epithelial cells with PA. Using Oil Red O staining, we demonstrated the presence and accumulation of lipids and triacylglycerides in STR, CPB and BAR-T cells as well as the EAC cell lines, OE33 and FLO1 (Figure 3a). The nature of the Oil Red O stain is based on hydrophobic interactions and association with lipids. Its general presence can also be interpreted as detection of lipids based on fatty acid synthesis not only uptake. LipidSpot^TM^ is a fluorogenic neutral lipid stain that allows the detection of lipid droplets with minimal background staining. Compared to the Oil Red O detection, lipid accumulation with LipidSpot^TM^ was mainly detected upon PA stimulation for 90 min compared to the BSA control, with a stronger baseline signal in the BE and EAC cell lines (Figure 3b), which was significantly enhanced in STR and FLO1 cells upon PA stimulation (Figure 3c). We further confirmed uptake of palmitic acids, which has a 16-carbon backbone, by using green-fluorescent fatty acid, BODIPY™ FL C_16_ (Appendix A). Co-localization of neutral lipids and mitochondria can be detected with BODIPY 493/503, a tracer for oil and other nonpolar lipids, and the labeling of mitochondria with MitoTracker (Appendix A). Furthermore, we established that the co-localization of 30 µM PA upon labeling with BODIPY 493/503 and mitochondria is disrupted by the etomoxir (Appendix A), indicating a halt of CPT1A-mediated lipid transport into the mitochondria. 

Together these experiments highlight that lipids such as PA can be taken up by esophageal epithelial cells and that the accumulated lipid droplets can target the mitochondria.

### 2.3. Palmitic Acid Promotes Esophageal Cell Proliferation and Colony Formation in BE and EAC Cell Lines

Next, we analyzed the functional consequences of lipid uptake by assessing cell viability and proliferation in the presence of 30 µM PA. While we noted no effect on proliferation for STR cells, STR B/A, which are pH4/bile salt-tolerant, showed significantly increased proliferation upon PA-treatment compared to control after 72 h (Figure 4a). Similarly, at 72 h, we demonstrated a trend and a significant increase in proliferation for BAR-T and CPA, respectively (Figure 4b). Both EAC cell lines, OE33 and FLO1 showed significant growth advantages at 48 h with PA stimulation (Figure 4c). We assessed the influence of carnitine as a stimulator of FAO [43] and the inhibitor etomoxir on the BE cell lines, CPA and BAR-T, since we observed a significant stimulation of proliferation in CPA but not BAR-T in the presence of PA. Both cell lines showed no increase in proliferation when FAO was stimulated with carnitine; however, both cell lines showed a significant proliferation decrease in the presence of etomoxir (Appendix A), demonstrating that CPT1 is the rate-limiting factor to the observed proliferation. To evaluate the consequences of etomoxir on cell growth in a physiological three-dimensional context in which the epithelial tissue architecture is preserved, we grew STR as a normal control, and CPA and BAR-T as well as OE33 as a tumorigenic representative as spheroids in 2% Matrigel. Treatment with etomoxir resulted in a dose-dependent overall decrease in spheroid size and number (Appendix A). However, when 1st generation spheroids were dispersed and seeded for 2nd generation spheroids in an attempt to assess the self-renewal capacities of cell lines along the BE tumorigenesis spectrum, we observed that normal STR controls could not form second generation spheroids in the presence of 100 nM etomoxir. Contrary, BE and OE33 spheres remained large in size (Appendix A). This could indicate an enhanced dependence on CPT1A-mediated FAO in the acquisition of stem-like features.

We therefore queried the capacity of self-renewal and stemness in the presence of PA and with CPT1A inhibition using the classic colony formation assay (Figure 4d–f). While BE and EAC cell lines formed colonies, the normal control STR and the STR bile/acid-tolerant cells did not form measurable colonies. In the presence of 30 µM PA, colony formation was stimulated significantly in both BE cell lines, CPA and BAR-T, and OE33 cells, but only a trend was observed for FLO1. Addition of etomoxir reduced the PA effect in all cell lines.

These observations demonstrate that palmitic acid can induce a growth advantage in BE and EAC cells.

### 2.4. IL1β and IL8 Expression Is Induced by PA Stimulation in the Early Stages of BE Pathology

Obesity is associated with low-grade inflammation and elevated levels of circulating saturated fatty acids, which trigger inflammatory responses through an increase in pro-inflammatory cytokines [44]. Relevant to the BE pathology are IL1β and IL8 which have been shown to be increased during the stepwise progression from normal through Barrett’s epithelium to adenocarcinoma [45]. We therefore assessed the expression of hIL1B and IL8 in the context of CPT1A expression upon PA stimulation and inhibition with etomoxir. Based on qRT-PCR data, hIL1B expression was significantly enhanced in STR B/A and BAR-T and FLO1 upon stimulation with PA but only reversed by CPT1A inhibition in BAR-T cells (Figure 5a). IL8 was significantly increased in STR, STR B/A, CPA and BAR-T. However, an upregulation was also observed in STR and STR B/A with the inhibition of CPT1A using etomoxir (Figure 5b). In comparison, CPT1A expression was significantly induced by PA in all cell lines but OE33. The increase in CPT1A with etomoxir corresponds with the earlier described increase in protein (Figure 5c, Appendix A). In summary, these data indicate that pro-inflammatory signaling correlated with FFAs and FAO may play a role during the early stages of the BE sequence.

### 2.5. CPT1A Is Upregulated by a High Fat Diet in Mice and Correlates with an Increase in Ki67-Positive Cells in Inflamed Human BE and EAC

To confirm the correlations between CPT1A expression, inflammation and proliferation, and their relevance to the human disease, we performed immunofluorescence staining on normal esophageal, BE, EAC and gastric control tissues. We observed the increase of CPT1A in the BE, EAC and glandular tissues of the stomach as described earlier (Figure 1) to be associated with an increase in signal for COX2, a central node for inflammatory signaling, and the proliferation marker Ki67 (Figure 6a). COX2 expression was mostly limited to the stromal compartment in BE tissues but was expressed in EAC tumor cells (Figure 6a). DCLK1 is a progenitor marker involved in the progression of Barrett’s tumorigenesis [46], which can be reduced with EAC treatment [47]. Analyzing human BE tissues, we aimed to detect DCLK1, CPT1A, COX2 and the keratin K8 at the junction of normal mucosa adjacent to BE lesions using multiplex immunofluorescence. We showed CPT1A to be elevated in DCLK1-positive cells. COX2 expression was mostly restricted to the stromal tissue (Figure 6a).

To investigate the consequences of a high fat diet on CPT1A expression in vivo, C57BL/6J mice went through a 24-week regimen of low fat calorie-matched diet and a fat diet (HFD). HFD-fed mice showed enlargement of the first cardia gland region in 17 of the 24 samples compared to 3 of the 12 LFD mice (*p* < 0.005, Appendix A) and increased signal for Alcian Blue indicating a higher amount of mucin-producing goblet cells (Appendix A). Analysis of stromal markers showed signs of inflammation as exemplified by higher pStat3 (data not shown). Inflammation has been shown to induce a higher expression of DCLK1 cells [48], which we found to be highly expressed in at the squamous-columnar junction of the HFD mice in conjunction with the stem cell marker Lgr5 and the proliferation marker Ki67. The morphology of DCKL1-positive cells resembles that of Tuft cells as highlighted with a higher magnification image (white arrows) In addition, the expression of CPT1A appeared to be elevated in a subset of cardiac cells at the squamous-columnar junction and present in the crypt regions of the stomach (Appendix A). 

Currently, the most accepted mouse model of BE is based on the targeted overexpression of the pro-inflammatory cytokine IL1β in the esophagus and forestomach [49]. When exposed to deoxycholate acid in the drinking water to mimic gastroesophageal reflux through the presence of bile, these mice phenocopied the sequence of esophagitis, Barrett-like metaplasia and EAC comparable to the human disease. Lgr5+ cells were increased at the gastric cardia and positive for DCLK1 possibly identifying them as BE progenitor cells. We placed these mice on a high fat diet, control low fat diet (matched calories) and combined treatments with DCA drinking water. At the time of tissue collection, all mice had developed metaplasia or EAC, many with additional squamous cell carcinoma. Using multiplex immunofluorescence staining we detected high signals for CPT1A, DCLK1 and COX2 in K8-positive BE-like regions, especially in mice which received the combination treatment (Figure 6b).

## 3. Discussion

Fatty acid oxidation has emerged as a new important pathway in tumor progression [20,50]. The association between visceral obesity and BE demonstrates a direct relationship between high fat diets and an increased risk for esophageal cancer [1,51,52,53]. At the same time, the mechanism of FFA potentially resulting in metaplastic transformation has not yet been thoroughly explored. Prolonged saturated fatty acid-rich diets induce chronic inflammation [54]. Additionally, the saturated fat found in a HFD can induce bile secretion into the intestine altering the bile acid pool and thereby the microbiota [55,56]. Both are contributing factors to BE tumorigenesis. In esophageal cancer, the role of fatty acid synthase (FASN) has been extensively described [57] and even proposed as a potential therapeutic target [58]. Fatty acid synthase is a rate-limiting enzyme for de novo lipid synthesis and is consistently found to be increased in multiple types of cancer [59], where it is thought to regulate stemness [60,61].

While FASN is a key enzyme of lipid synthesis, CPT1A plays a critical role in FAO. The role of CPT1A-mediated FAO in the context of promoting cell growth has been described in other cancers [62,63], but not in esophageal cancer. We show an increased CPT1A expression in human tissue specimen along the sequence of Barrett’s tumorigenesis (Figure 1). Recent genomic data support additional classification and division into BE subtypes, one of which is dictated predominantly by the gene and DNA methylation signature associated with elevated ATP synthesis and fatty acid oxidation [64]. We show that BE and EAC cell lines are more likely to express CPT1A than normal esophageal squamous cell lines (Figure 2), and additionally, that CPT1A is the predominant isoform amongst the family of CPT1s (Appendix A). It has been reported that the receptors which bind free fatty acids may already play a role in early BE pathogenesis: FFAR3 showed the highest and FFAR4 exhibited the lowest expression in all esophageal samples and significantly correlated with the severity of microscopic damage in GERD. Higher relative expression of FFAR1 and FFAR2 and significantly higher expression of FFAR3 was observed in patients with GERD compared to healthy controls [65].

Inflammation is a key step in the progression of GERD [14,15,16,17,18,19,20,21]. PA has been shown to induce pro-inflammatory cytokines, such as IL6, IL1β and TNF, NF-KB [66]. Stimulation with PA induced IL1B and IL8 in our experimental BE cells lines but did not show an effect in the EAC cells (Figure 5) supporting the importance of fatty acid exposure in the early steps of BE pathogenesis. The upregulation of pro-inflammatory cytokines like IL1B has been shown to not only cause inflammation but also cause Barrett’s-like metaplasia in a mouse model of IL1β expression in the esophagus and forestomach [49,67]. In our hands, these mice develop esophageal squamous cell carcinoma and adenocarcinoma early on (9 month), even in the absence of the deoxycholate challenge as in the original model [49]. We believe that the process of establishing a clean colony altered the microbiota contributing to an accelerated phenotype. IL1β mice developed esophageal dysplasia and tumors more rapidly when fed a HFD than mice fed the control diet as reported by the Quante’s group [13], independent of body weight. These findings were recapitulated in our colony as mice on a HFD did not gain weight. More so, the acceleration of dysplasia by the HFD in the IL1β mice was associated with a shift in the gut microbiota [13] supporting our hypothesis that tumor formation in this model in our hands was accelerated due to the eradication of microbiome during rederivation.

Interestingly, it has been reported in humans that the presence of visceral adipose tissue is associated with the risk of presenting long segment Barrett’s esophagus, even in subjects which did not appear obese [68]. This indicates that diet-associated inflammation alone could contribute to poor outcome. The cytokine IL8 was elevated in IL1β-HFD mice and it was shown that organoids from mice produced increased levels of Il8 upon stimulation with serum from obese human patients [13].

Mechanistically, LGR5+ gastric cardia stem cells present in the murine BE-like lesion were suggestive of gastric progenitors arising under tumor-promoting inflammatory conditions [49]. Additionally, increased numbers of DCLK1+ cells were found during the expansion of the first gastric gland considered to be at the heart of BE tumorigenesis [69]. Serum DCLK1 has been shown to be a potential marker [48] as it can be detected early in BE and in EAC [49,64]. DCLK1 detection by immunofluorescence staining allowed a direct delineation of the BE lesion adjacent to the normal squamous tissues and appeared to co-localize with increased CPT1A signal in the same region in human tissue (Figure 6). We observed the same correlation in IL1β mice, especially upon combination treatment with deoxycholic acid and HFD, but overall expression was elevated in the background of IL1β-induced inflammation and tumorigenesis. We therefore assessed C57/B6 mice on a HFD and found CPT1A to be highly expressed at the squamous-columnar junction of the HFD mice in conjunction with the stem cell marker LGR5 and the proliferation marker Ki67. This finding suggests a more direct link between HFD and CPT1A upregulation in the absence of IL1β-induced inflammation. Based on the region of reference and the morphology, we believe that the DCLK1-positive cells could potentially resemble Tuft cells (Appendix A). As described by Middelhoff et al. [70], DCLK1-expressing Tuft cells represent a fifth epithelial cell lineage and while not stem cells per se, they are required for intestinal regeneration and have been shown to be expanded during chronic injury and early tumorigenesis [71]. Interestingly, they are also considered to be chemosensory and express taste cell-specific GTP binding protein a-gustducin and the cation channel transient receptor potential melastatin subtype 5 (TRPM5) [72]. As described above, DCLK1 has been proposed to be a marker in human BE and EAC tissues [49,64], however, the positively stained cells appear different from the mice Tuft cells and it is unclear if they retain similar function or are a true feature of the human disease.

Elevated FAO has been shown to ensure the energy resources for the extreme environment alteration in cancer and their cancer stem cells. Hence, CPT1 has been considered a critical accelerator of FAO, and shown to promote breast cancer stemness and chemoresistance [73]. Targeting CPT1A-mediated FAO has also been shown to sensitize nasopharyngeal carcinoma to radiation therapy [74]. The authors of that study also demonstrated that CPT1A binds to RAB14 and promotes fatty acid trafficking using BODIPY 493/503-labled lipid droplets and BODIPY C16. Our experiments identified the uptake of palmitic acid by esophageal cells and the trafficking to mitochondria as shown by co-localization of labeled lipid fractions and mitochondria (Figure 3, Appendix A). Differences in lipid supply have been shown to be dependent on signaling activation: increased lipid uptake for example was observed in cells transformed by oncogenic HRAS, whereas cells transformed by constitutively active AKT instead showed increased de novo synthesis [75]. When assessing the functional consequences of elevated supplies of free fatty acids such as palmitic acid, a main component of high fat or Western diets, we observed enhanced proliferation and colony formation in BE and EAC cells most likely associated with stemness (Figure 4). More so, when grown in three-dimensional spheroids (Appendix A), we observed a dose-dependent effect of CPT1A-inhibtion using etomoxir on spheroid growth. CPT1A has been shown to protect colorectal tumor cells from anoikis [27] and supports BE and EAC spheroid growth, which is inhibited upon disruption of FAO. Mana et al. [76] in a study focusing on intestinal stemness and tumorigenicity in response to high-fat diet driven FAO tested if FAO inhibition using etomoxir would attenuate the HFD-induced increase in organoid formation, which were isolated from murine models on HFD. When isolated crypts from control and HFD-fed mice were grown in organoid assays with increasing concentrations of etomoxir, etomoxir treatment had no effect on the ability of control crypts to form organoids, but reduced clonogenicity of HFD-crypts in a dose-dependent manner. Interestingly, the same study suggested that early intestinal tumors arising from HFD induced intestinal stem cells are highly sensitive to FAO inhibition. This finding provides a potential context for our observation that spheroids established from normal and BE cell lines had a smaller diameter than EAC spheroids in the presence of etomoxir and that upon dissociation and growth as 2nd generation spheroids, normal cells were unable to form spheroids but all other cells were unaffected by the treatment. We show a similar correlation in colony formation assays, in which palmitic acid stimulated an increase in colony numbers and etomoxir-mediated FOA disruption attenuated the observed increase.

While we did not identify a molecular mechanism for the palmitic acid-mediated functional alternations, there is emerging evidence that palmitic acid, one of the saturated fatty acids in HFD diets, and CPT1A are involved in intracellular signaling associated with cancer development beyond being an energy source [24,27,28]. Lipids themselves can play a role in transcriptional regulation, e.g., PPARs, a family of lipid-activated nuclear receptors [77]. A high fat diet has been shown to induce PPAR in intestinal stem cells and non-intestinal progenitors. More specifically, high fat diet in this study enhanced the number of LGR5+ intestinal stem cells [78,79]. Additionally, PPARs appear to regulate the expression of the self-renewal gene Sox2 at the transcriptional level [80]. 

Together, this suggests that lipids and lipid-bound nuclear receptors may directly regulate the expression of stem cell specific genes. Interestingly, Sox2 has been shown to cooperate with inflammation-mediated Stat3 activation in the malignant transformation of foregut basal progenitor cells [81,82] contributing to EAC. More relevant to our study, inhibition of the PPAR/fatty acid synthesis axis potently suppresses esophageal adenocarcinoma as shown by cell viability, identification of transcription factors, and the master regulation of fatty acid synthesis in EAC [83]. Although we have not explored the fatty acid synthesis side, the fact that CPT1A expression increases in BE and EAC and subsets of BE are characterized by a FAO phenotype, suggests that CPT1A and FAO are one of the main roads to cancer. Supporting this is the fact that some of the most highly co-expressed genes of CPT1A are FAO genes, ACOX1 and ACSS1, and PPARa in EAC and other gastrointestinal cancers (Appendix A) but FASN does not show the same trend.

In summary, we have shown in this study that fatty acid metabolism, i.e., palmitic acid upon uptake by esophageal cells provides a growth advantage and correlates with the acquisition of progenitor marker expression characteristic of Barrett’s tumorigenesis.

## 4. Materials and Methods

### 4.1. Cell Culture

Human esophageal epithelial cells, STR, were cultured as previously described. In brief, Keratinocyte-SFM medium was supplemented with 1 ng/mL Epidermal Growth Factor, 0.05 mg/mL Bovine Pituitary Extract, and 1% penicillin streptomycin antibiotics (Gibco™, for Life Technologies, Inc., Carlsbad, CA, USA). Bile/acid-tolerant STR cells were established by repeated exposure to a 100 μM of cocktail of bile salts (20 μM of each, deoxycholic acid (MP Biomedicals, Santa Ana, CA, USA), glycocholic acid, taurocholic acid, sodium glycodeoxycholate and sodium glycochenodeoxycholate (all Sigma, St. Louis, MO, USA) as reported) [34].The non-neoplastic Barrett’s epithelial cell lines CPA, CPB, BAR-T and BAR-10T cells were isolated from non-dysplastic metaplasia (kind gift from Dr. Rhonda Souza, Baylor Scott White) [35], and were cultured in Epithelial Cell Medium-2 supplemented with 5% epithelial cell growth supplement-2 (EpiCGS-2) and 5% penicillin/streptomycin antibiotics (ScienceCell™ Research Laboratories, Carlsbad, CA, USA). The esophageal adenocarcinoma cell lines, FLO1, OE33 and OE19 (kind gift of Dr. El-Rifai, University of Miami), were grown in RPMI with 10% FBS and 1% penicillin/streptomycin antibiotics (ThermoFisher Scientific, Waltham, MA, USA). All cells were incubated at 37 °C with 5% CO_2_.

### 4.2. Human Tissues

Normal, normal adjacent and esophageal adenocarcinoma were procured from the Cooperative Human Tissue Network (CHTN) as de-identified biospecimen (Exempt study upon review by the Institutional Review Board of the University of Florida). 5-micron sections of archived FFPE-Barrett’s esophagus biopsies with metaplasia and a transition to normal were provided and pathologically reviewed by Dr. Qureshi.

### 4.3. Animals

All mouse studies and breeding were carried out under the approval of IACUC of the University of Central Florida (protocol codes #20200032 and #17-54).

C57BL/6J mice were placed on a high fat diet, TD.96121 (42% adjusted calories from fat with 1.25% cholesterol) and matching low fat control diet TD.08485 for 24 weeks. The weight was monitored over the course of the experiment and mice were sacrificed at 24 weeks. 

Human IL1β transgenic mice were a kind gift of Dr. Timothy Wang (Columbia University) and generated by targeting expression of hIL1β to the esophagus using the Epstein Bar virus promoter (L2) [50]. Upon receipt at the University of Central Florida, the mice were treated with the antibiotic Baytril to establish re-derived clean litters in the care of foster mothers and continued for breeding. L2-IL1B, Tg(ED-L2-IL1RN/IL1B) were placed on drinking water containing bile acids (0.3% DCA, pH 7.0) for three months. Additional experimental groups were started on a high fat diet (custom Teklad diet TD.88137, 42% adjusted calories from fat, 1.5 g cholesterol/kg) and control low fat diet TD.08485 (Envigo, Madison, WI, USA), both irradiated, at 8 weeks old, some in combination with the DCA drinking water. Control mice without any treatment were sacrificed for tissue analysis at the same end points.

Mouse tissues from the esophagus and forestomach were isolated and fixed in formalin, paraffin-embedded, and then cut and stained with H&E (Hematoxylin and eosin).

### 4.4. Histology

Five-micron sections were applied to Probe-on Superfrost Plus slides (Fisher Scientific, Pittsburgh, PA, USA). Slides were stained with hematoxylin and eosin, and images were captured on a Zeiss microscope with a Zeiss Mrc5 camera and acquired with Zen blue 3.1 (Carl Zeiss Microscopy, Thornwood, NY, USA).

### 4.5. Conjugation of Palmitic Acid

A palmitic acid (Tocris, Bristol, UK) stock solution 100 mM was made in 10 mL of a 100 mM NaOH. The 0.17 mM BSA solution was made by dissolving BSA (Sigma-Aldrich, St. Louis, MO, USA) in 150 mM NaCl. Palmitic acid was conjugated with BSA by diluting 1/100 of the stock solution in 0.17 mM BSA in PBS and equilibrating the pH at 7.4. Conjugated PA was aliquoted and frozen for storage prior use.

### 4.6. Spheroid Cultures

Cells were seeded at 1000 cells per mL in their respective growth medium with 2% matrigel on low adherence plates (Corning, Corning, NY, USA). Control BSA treated cells were grown with 34 µM BSA (Sigma-Aldrich, St. Louis, MO, USA) final concentration, and treatments contained 0.5 µM of L-carnitine with BSA or 30 µM final concentration BSA-conjugated Palmitic Acid (Tocris, Bristol, UK). Etomoxir (Tocris, Bristol, UK) was used at 10 and 100 µM in the spheroid growth medium. Spheroid cultures were maintained for 2 weeks; the treatments were added on day 0 and renewed on day 7. Spheroid diameter was determined before harvest by counting 10 separate 100× fields with ImageJ. After imaging and counting, 1st generation spheroids were dissociated using Trypsin and 1000 cells seeded with 2% matrigel on low adherence plates for 2nd generation spheroids. These were maintained for another 3 weeks in the absence of new etomoxir treatment but with renewing of growth media. At the end of the incubation period, spheroid diameter was determined after imaging.

### 4.7. Proliferation Assay

To determine cell viability and proliferation, 2000 cells/well were seeded in a 96 well plate. CellTiter-Blue^®^ reagent (Promega, Madison, WI, USA) was added to the wells and incubated for an hour prior to recording fluorescence at 560/590 nm using a Spectramax i3x Multi-Mode Detection system (Molecular Devices, San Jose, CA, USA) following the manufacturer’s instructions. 

### 4.8. Colony Formation Assay

Cells were seeded at 1000 cells per well in a 6-well plate (Corning, Corning, NY, USA) and grown for 10 days prior to fixation in Methanol:Acetone 1:1 for 10 min at −20 °C. Cells were stained with 0.5% crystal violet in methanol. Multiple pictures per well were taken and the size of the colonies was quantified using ImageJ by counting pixel density in each well.

### 4.9. qRT-PCR

Total RNA was extracted and purified from STR and BAR-T cells after each treatment via phenol and guanidine thiocyanate extraction using the miRNeasy^®^ Mini Kit (Qiagen, Santa Clara, CA, USA) according to the manufacturer’s protocol. Reverse transcription was performed using QuantiTect^®^ Reverse Transcription Kit (Qiagen, Santa Clara, CA, USA). The subsequent Real time qRT-PCR was then performed using the SYBR^®^ Green PCR kit by Qiagen. The genes Rpl13A and Eif3d were used as internal controls. Primer sequences were selected as followed (Integrated DNA Technologies, Coralville, IA, USA): CPT1A, forward 5′-ATC AAT CGG ACT CTG GAA ACG G-3′, reverse 5′-TCA GGG AGT AGC GCA TGG T-3′; CPT1B. forward 5′-GCG CCC CTT GTT GGA TGA T-3′, reverse 5′-CCA CCA TGA CTT GAG CAC CAG-3′; CPT1C, forward 5′-GGA CTG ATG GAG AAG ATC AAA GA -3′, reverse 5′-CAC AAA CAC GAG GCA AAC AG-3′; hIL1B, forward 5′-GGA GAT TCG TAG CTG GAT GC-3′, reverse 5′-GAG CTC GCC AGT GAA ATG AT-3′; hIL8, forward 5′-CCT GAT TTC TGC AGC TCT GTG-3′, reverse 5′-CCA GAC AGA GCT CTC TTC CAT-3′.

### 4.10. Western Blot

Cells were lysed 30 min post treatment using IP Lysis buffer (150 mM NaCl, 50 mM Tris, pH 8.0, 1% Triton x-100, 1% NP-40) supplemented with a cOmplete™ EDTA-free protease Inhibitor Cocktail tablet. Protein concentration was determined using the Pierce™ BCA Protein Assay Kit (Thermo Scientific, Waltham, MA, USA). Subsequently, protein samples were separated on a 10% acrylamide gel via SDS PAGE. Gels were transferred to a PVDF membrane. Proteins were visualized using primary antibodies and HRP-conjugated secondary antibodies.

### 4.11. Immunofluorescence

Formalin-fixed paraffin embedded spheroids or human tissues were sectioned at 5 µm, deparaffinized and heated in 1x TE buffer in a pressure cooker for 12 min for antigen retrieval. Samples were blocked in PBS containing 5% Bovine Serum Albumin (BSA; Sigma-Aldrich, Saint-Louis, MO, USA) for 1 hour before incubation with primary antibodies in PBS-BSA overnight at 4 °C. Tissues were then rinsed in PBS and incubated with secondary antibodies in PBS-BSA for 1 h at room temperature. Finally, the sections were mounted with DAPI Fluoromount-G^®^ mounting medium containing DAPI (SouthernBiotech, Birmingham, AL, USA). Pictures were taken on a Zeiss microscope Imager.M2, equipped with an Axicam M506 mono camera using Zen blue 3.1 software (Carl Zeiss Microscopy, Thornwood, NY, USA). 

### 4.12. Tyramide-Based Multiplex Immunofluorescence Staining

5 µM formalin-fixed, paraffin-embedded (FFPE) tissues sections were deparaffinized with xylenes, rehydrated, and incubated for 13 min in 1x TE in a Cuisineart pressure cooker. After cooling down for 40 min, slides were incubated with the first primary antibody (Anti-CPT1A antibody clone EPR21843-71-1C from Abcam #ab220789, 1:200 diluted) overnight at room temperature in 1x PBS/5% bovine serum albumin (BSA). Slides were washed with 1x PBS and incubated for 15 min with ImmPRESS HRP Horse Anti-Rabbit IgG Polymer Detection Kit from Vector Laboratories (order #MP-7401). Slides were washed again with 1x PBS and incubated with fluorescent dye conjugated tyramides from Biotium (Fremont, CA, USA), i.e., FITC (order #96018; 0.5 mg/200 µL): 1:800 dilution, and CF660R (#92195; 0.5 mg/200 µL): 1:250. Dilutions were made in a 0.1 M borate buffer with hydrogen peroxide added immediately before staining to a final concentration of 0.003%. Slides were incubated for 15–25 min at room temperature with a tyramide and then washed in 1x PBS. For the next round of staining, the antibodies were removed by heating in a microwave: slides were incubated in 1x TE buffer and the buffer was brought to boiling. This was repeated three more times over the next 30 min and then the slides were cooled down to room temperature. Then, the second primary antibody (Anti-DCLK1 clone EPR6085 from Millipore/Sigma #MABN1115, 1:350 diluted) was added for overnight incubation. For a 4-antibody stain the sequence started with FITC-tyramide, followed by CF660R-tyramide. The last two antibodies (Anti-KRT8 cloneKs8.7 from Santa Cruz Biotechnology #sc-101459, 1:70 diluted; and Anti-Cox2/PTGS2 clone D5H5 from Cell Signaling Technology #12282, 1:100 diluted) were detected with a mix of Horse Anti-Mouse IgG Antibody (H+L), CY3 (Vector Laboratories order #CY-2300) and Horse Anti-Rabbit IgG Antibody (H+L), Biotinylated (Vector Laboratories order #BA-1100), both 1:400 diluted. After 15 min slides were washed with 1x PBS and incubated with Streptavidin-Pacific Orange conjugate (Thermofisher order #S32365; 1:400 diluted) for 15 min. After final washes, slides were mounted with Southern Biotech’s DAPI Fluoromount-G (order #0100-20) and imaged using a Leica SP5 confocal microscope.

### 4.13. Lipid Detection Using Oil Red O Staining, BODIPY and LipoSpot 610

50,000 cells were seeded on 4-chamber glass slides in 1mL of medium and incubated for 24 h to allow for attachment. On the following day, medium was exchange and 34 µM BSA was added to the control cells, the other wells were treated with BSA-conjugated Palmitic acid at 30 µM. After 90 min of treatment, the medium was removed followed by a 1x PBS wash. The cells were fixed in 10% formalin for 5 min, then incubated in isopropanol for 5 min, followed by incubation with 0.5% Alfa Aesar Oil Red O (Fisher Scientific, Waltham, MA, USA) diluted 60/40 in water and filtered. Slides were rinsed, mounted with Fluoromount-G^®^ (SouthernBiotech, Birmingham, AL, USA) and visualized. 

For fatty acid tracing, 50,000 cells were seeded on 4-chamber glass slides in 1mL of medium and incubated for 24 h to allow for attachment. On the following day, cells were incubated with complete medium supplemented with either 34 µM BSA or 30 µM of BSA-conjugated Palmitic acid for 90 min, the medium was removed followed by a 1x PBS wash. Cells were then fixed with 10% formalin for 15 min, followed by permeabilization with 0.1% Triton X-100 for 20 min. LipidSpot^TM^ was added to the slides according to the manufacturer’s instructions (Biotium, Fremont, CA, USA) The slides were rinsed and mounted with DAPI Fluoromount-G^®^ mounting medium containing DAPI (SouthernBiotech, Birmingham, AL, USA). Random images of three biological replicates were taken per condition and positive cells counted per field and compared to total nuclei for quantification.

Alternatively, cells were incubated with complete medium containing 1 μM BODIPY 493/508 (Invitrogen, MA, USA) after PA and BSA treatment or directly with BODIPY C16 to allow the fluorescent fatty acids to metabolize. Mitochondria in this combination were labeled with 50 nM MitoTracker Red CMXRos (Invitrogen) for 30 min prior to imaging.

### 4.14. Biostatistical Analysis

In vitro and in vivo biostatistical analysis was performed using Prism version 6.00 for Mac. When comparing only two conditions to one another we used a Student’s *t*-test (Welch’s correction) to analyze statistical significance differences. When comparing multiple conditions to the untreated control or palmitic acid treatment, we used One-Way ANOVA (Dunnett’s correction) to analyze statistical significance differences. Statistical significance was set at *p* < 0.05. All experiments were done in triplicates with at least three biological replicates.

### 4.15. Dataset Analyses

Public GEO dataset GDS1321 and GSE34619 were queried for components of fatty acid metabolism during the progression from normal esophagus to esophageal adenocarcinoma. 

## 5. Conclusions

This study highlights the relationship between fatty acid metabolism and the acquisition of characteristics of Barrett’s tumorigenesis.

## Figures and Tables

**Figure 1 ijms-23-11745-f001:**
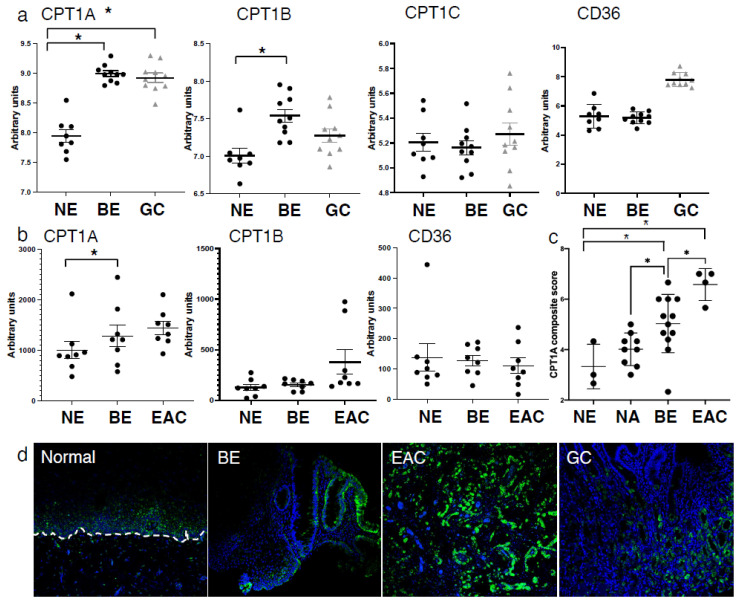
Expression of CPT1A is increased in the sequence from normal esophageal squamous epithelium to BE and EAC. (**a**) Comparison of CPT1 component expression and CD36 between normal esophagus (NE), Barrett’s esophagus (BE) and gastric cardia (GC) using the GSE34619 dataset (8 patients normal squamous, 10 with BE and 10 gastric cardia). * *p* < 0.05 Student’s *t*-test (**b**). Expression of CPT1A, CPT1B and CD36 in normal esophagus (NE), Barrett’s esophagus (BE) and esophageal adenocarcinoma (EAC) tissues in the publically available dataset GDS1321/GSE1420;8 patients with Barrett’s associated adenocarcinomas. (**c**) Composite score and quantification of immunofluorescent staining using antibody against CPT1A in normal esophagus (NE), Barrett’s esophagus (BE) and esophageal adenocarcinoma (EAC) and matching normal adjacent tissues (NA). (**d**) Representative images of CPT1A immunofluorescence staining. * *p* < 0.05, Student’s *t*-test.

**Figure 2 ijms-23-11745-f002:**
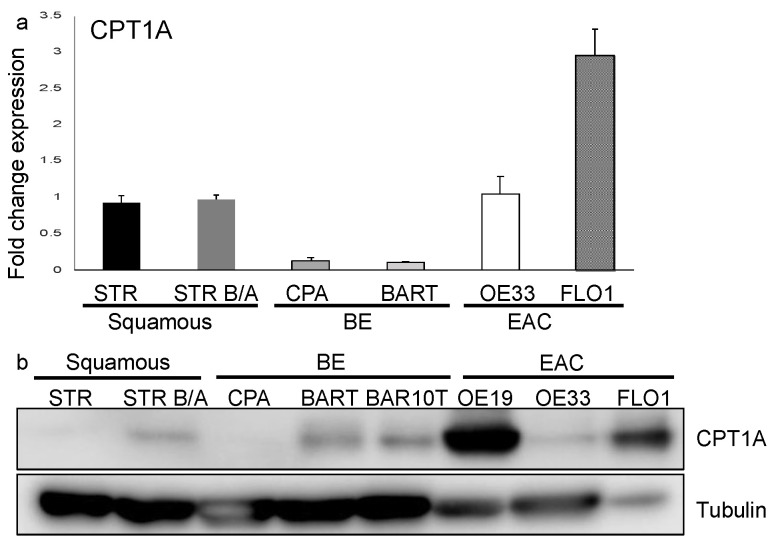
CPT1A is expressed in the esophagus and protein upregulated in the progression to EAC. (**a**) qRT-PCR with primers for CPT1A amplification was performed using RNA extracted from normal immortalized esophageal keratinocytes (STR), bile/acid-tolerant STRs, the Barrett’s esophagus cell lines, CPA and BAR-T, as well as esophageal adenocarcinoma cell lines (OE33 and FLO1) was evaluated. RNA expression is increased in EAC cells. (**b**) Protein was extracted from cell lysates and detected with antibody against CPT1A by Western Blot, showing an increased expression in bile/acid-tolerant STRs, BAR-T and BAR10T as well as EAC cell lines (OE19, OE33 and FLO1) compared to normal.

**Figure 3 ijms-23-11745-f003:**
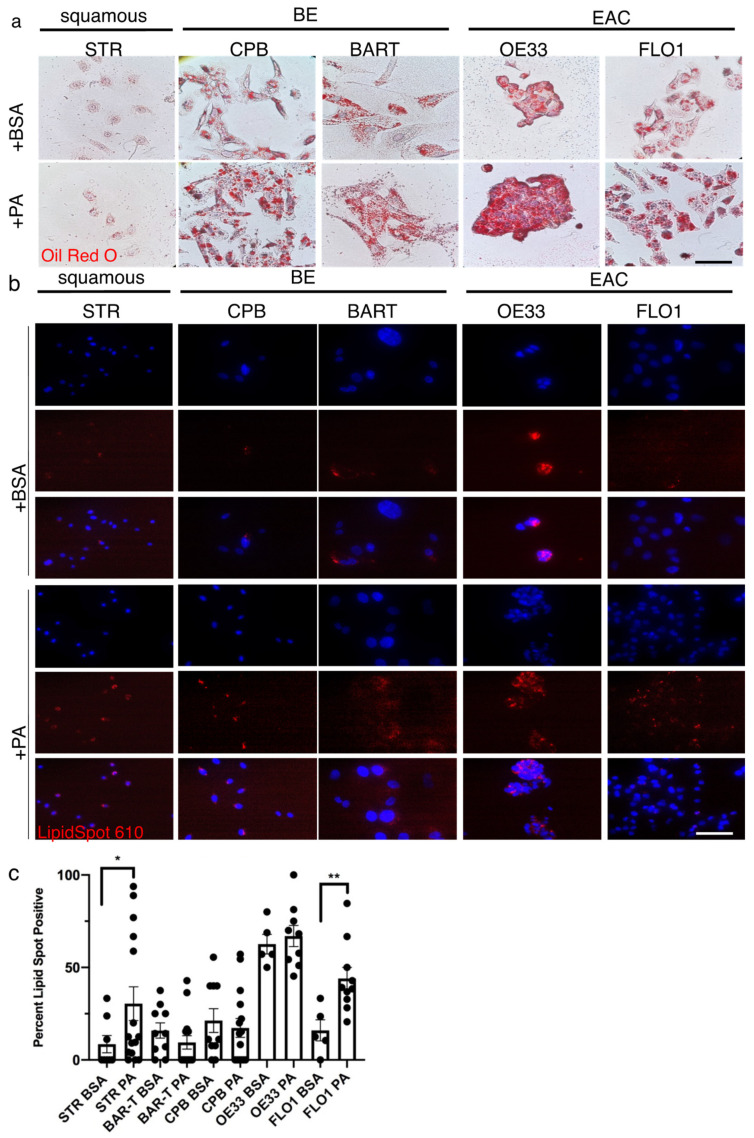
Palmitic acid uptake can be detected in esophageal epithelial cells. (**a**) The Oil Red O stain shows a general presence of lipids in STR, CPB and BAR-T cells as well as the EAC cell lines, OE33 and FLO1. (**b**) The fluorescent red signal using LipidSpot^TM^ for the detection of neutral lipids is observed in the BE cell lines, CPB and BAR-T, and EAC cells, OE33 and FLO1 after treatment with 30 µM palmitic acid for 90 min compared to BSA control, DAPI for nuclear stain. (**c**) Lipidspot-positive cells were counted per field and compared to total nuclei for quantification. * *p* < 0.05, ** *p* < 0.01, *t*-test with Welch’s correction.

**Figure 4 ijms-23-11745-f004:**
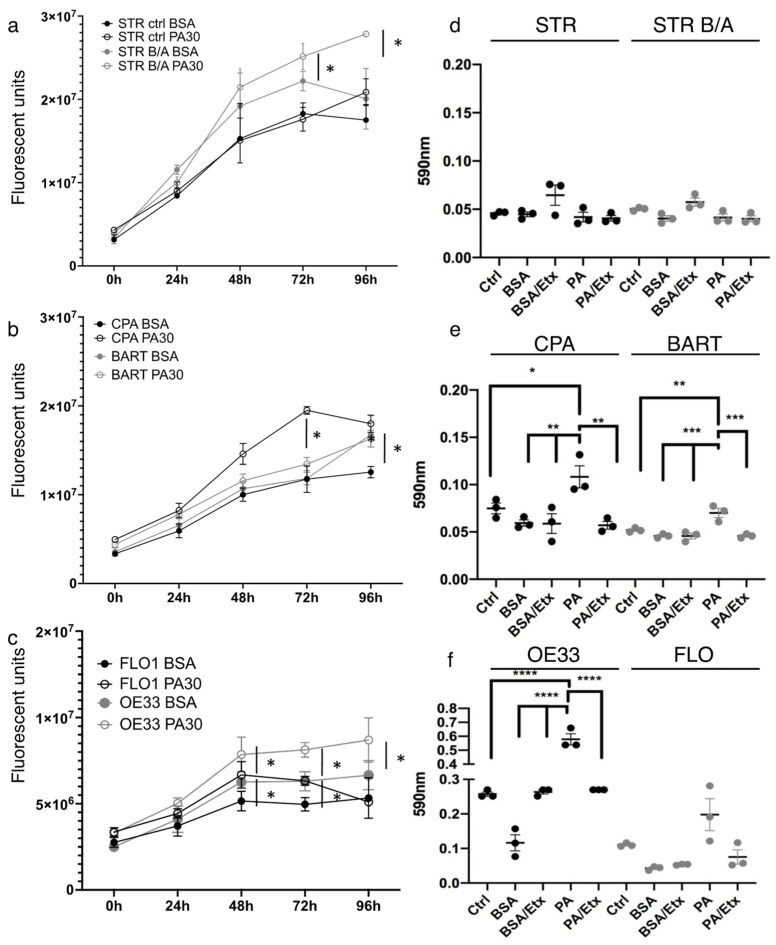
Palmitic acid promotes proliferation and increases the capacity for colony formation in EAC. Cell proliferation was determined using the CellTiter assay showing that stimulation with 30 µM palmitic assay increases growth significantly compared to BSA control in bile/acid-tolerant STR but not normal esophageal STR (**a**). (**b**) The proliferation of BE cell lines, CPA and BAR-T, is enhanced with 30 µM palmitic acid, so is cell growth in esophageal adenocarcinoma cell lines compared to BSA control (**c**). Crystal violet stain was quantified for colony formation of STR and STR B/A (**d**), BE (**e**) and EAC (**f**) cell lines. STR and STR B/A did not form measurable colonies, but colony formation is significantly increased in CPA, BAR-T and OE33 cells upon stimulation with 30 µM PA. There is a trend for augmented colony formation in the FLO cells, but it is not significant. * *p* < 0.05; ** *p* < 0.01; *** *p* < 0.001; **** *p* < 0.0001.

**Figure 5 ijms-23-11745-f005:**
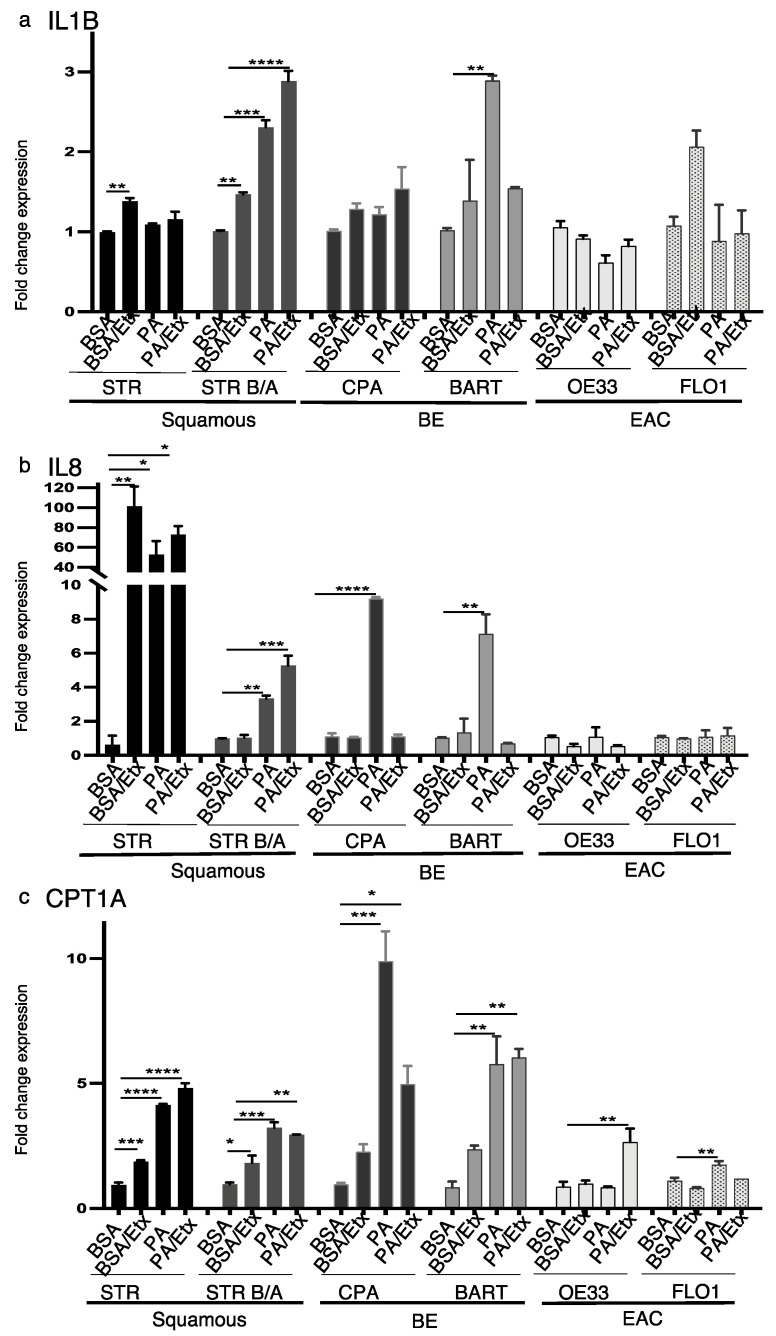
PA stimulates IL1B, IL8 and CPT1A and upregulation in the progression to EAC. RNA expression was determined by qRT-PCR with primers for the inflammatory cytokines IL1B and IL8 and CPT1A. (**a**) Palmitic acid increased the expression of IL1B in bile/acid-tolerant STRs and BAR-T cells. Etomoxir only inhibits PA-dependent increased expression in BAR-T. (**b**) IL8 expression is increased in a PA-dependent manner in BE cell lines and inhibited by etomoxir. (**c**) CPT1A is increased in response to PA stimulation in STR, CPA, and BAR-T cells. Etomoxir only reverses the increase in CPA cells. * *p* < 0.05, ** *p* < 0.01, *** *p* < 0.001, **** *p* < 0.0001, One-way ANOVA.

**Figure 6 ijms-23-11745-f006:**
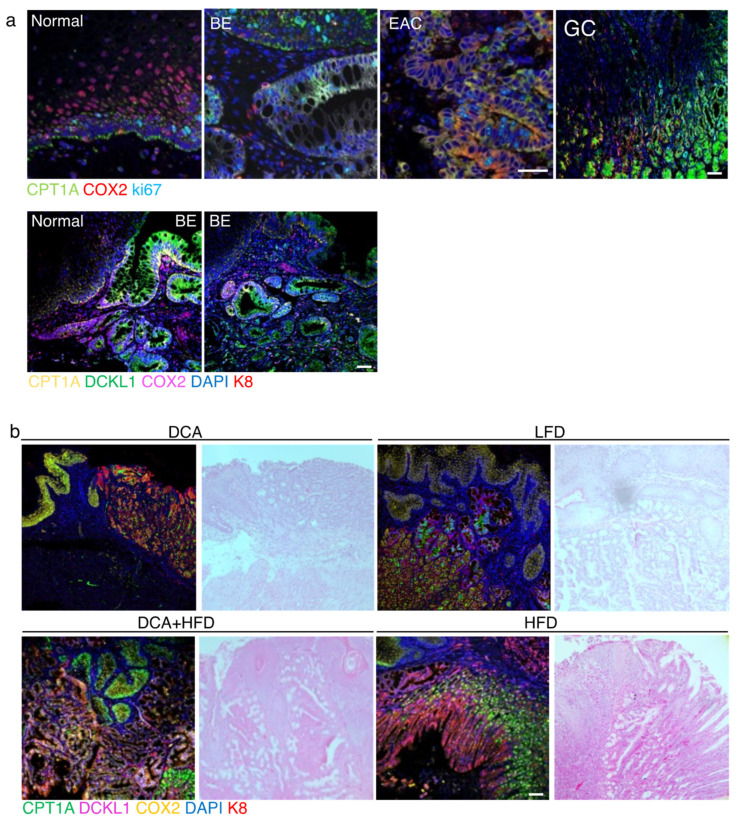
CPT1A upregulation correlates with an increase in COX2 and DCLK1 in human and mouse. (**a**) Human biospecimen from normal, Barrett’s (BE), esophageal adenocarcinoma (EAC) and gastric cardia were stained with antibody against CPT1A, COX2 and ki67. CPT1A and COX2 expression are increased in BE, EAC as is ki67. In biopsies capturing the junction of healthy squamous and BE, the increase in CPT1A and the stemness marker DCLK1 localize to the BE lesion, also identified with K8 using multiplex immunofluorescence staining. (**b**) ED-L2- IL1β mice were exposed to deoxycholate (DCA) in the drinking water, low fat diet (LFD) or high fat diet (HFD) alone and in combination with DCA. The combination treatment results in a high signal for the detection of CPT1A, DCLK1 and COX2 in K8-positive BE-like regions. Scale bar, 50 microns.

## Data Availability

Not applicable.

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
