# Peer review of "Augmented CPT1A Expression Is Associated with Proliferation and Colony Formation during Barrett’s Tumorigenesis"

_ijms, 2022, doi:10.3390/ijms231911745_

Round 1

Reviewer 1 Report

The article explores the relevance of CPT1 on the development of esophageal and gastric cancer. Although this enzyme was explored in other setting the study is relevant to understand the transformation mechanisms. Several issue however makes the paper preliminary or mine the robustness of reported findings. In particular may experiments lacks appropriate quantification, reporting of replicates done and statistical testing (see below for a list). In addition the reported link between FA and stemness in this

system is supported by histological co-occurrence and poorly detailed in vitro replating experiments. Please also discuss 30 uM PA as compared to free fatty acids in obese patients’ plasma.

Thus the paper is not suitable for publication at this time but submission of a revised version is strongly encouraged.

Major points

Data in Fig.3 are key in the paper but lacks quantification. Quantification and normalization for the cell number is mandatory to evaluate the extent of stain increase upon PA treatment. Moreover, red fluorescence in panel b is hard to appreciate, maybe better to show the two channels separately.

Figure 5 lacks statistical hypothesis testing and indications of how many times the experiments were repeated.

Figure 6 only shows representative images, blind scoring from an experienced pathologist and statistical testing are required.

Please improve the statistical testing, method description and discussion about ETx a spheroids as this the evidence linking PA and stemness. Expression of stemness related genes in function of FA treatments and ETX exposure is also required.

Figure S4 is of poor quality, lacks quantification and testing.

Minor points

Fig.1a and Fig1b please revise significance marks in left panel GC is misplaced under BE and in right panel GC seems significant but unlabeled. PLEASE ANALYZE AND REPORT ALL AGAINST NE CONTROL.

Fig.2 quantification in (a) misses some samples shown in (b), why?

Fig.4-d-e-f the Y scale used makes differences hard to appreciate apart for OE33, why not to use log scale?

Author Response

We like to thank the reviewer for the suggestions and guidance on how to improve this manuscript!

Major points

  1. Q: Data in Fig.3 are key in the paper but lacks quantification. Quantification and normalization for the cell number is mandatory to evaluate the extent of stain increase upon PA treatment. Moreover, red fluorescence in panel b is hard to appreciate, maybe better to show the two channels separately.

R: The intent of these experiments was to demonstrate that esophageal epithelial cells can take up free fatty acids/lipids such as palmitic acid, which to our knowledge has not been reported before. We state that we can detect the accumulation of lipids with Oil Red O in all cell lines, but detection using the LipidSpotTM shows a more selective increase, especially in BE and EAC cells upon PA stimulation. We therefore focused on the quantification in these experiments and provide a new figure 3 with images for separated channels in panel b and a graph for the quantification, panel c (page 6). Statistics were performed using a t-test with Welch’s correction, and the p-values are indicated in the figure. The materials and methods were expanded to include the quantification (page 17)

  1. Q: Figure 5 lacks statistical hypothesis testing and indications of how many times the experiments were repeated.

R: Thank you for pointing to the lack of statistics. We analyzed all three biological replicates combined for the new data in Figure 5 containing analysis using a One-Way ANOVA. Upon including all replicates, data for IL8 showed an induction upon treatment in STR cells, while the data points for this gene in the other cell lines and for IL1B and CPT1A did not change. We modified the figure (page 9) and the legend to reflect statistical differences as assessed with a One-Way ANOVA.

  1. Q: Figure 6 only shows representative images, blind scoring from an experienced pathologist and statistical testing are required.

R: We agree with the reviewer that we only show representative images and no quantification of statistical testing for the multiplex immunofluorescence stainings. The sections themselves were selected based on review of the tissue pathology for staining yet the 4 color immunofluorescence is difficult to quantify manually. Our assessment with restricted antibody use (as in Figure 1) generates a composite score which combines values for the number of positive cells per section imaged, staining intensity, and co-localization (e.g., same cell or nuclear vs cytoplasmic /membranous). For four colors, this is usually done automatically to generate values ready for statistical analysis, and while we recently acquired the MACSima imaging platform for robustness of quantification, not all antibodies and protocols have been tested for this type of fully automated fluorescence microscopy of multiple samples. We therefore do not make a statement to specific cells being positive for all 4 markers, but rather the association of each within the context of the sample.

  1. Q: Please improve the statistical testing, method description and discussion about ETx a spheroids as this the evidence linking PA and stemness. Expression of stemness related genes in function of FA treatments and ETX exposure is also required.

R: We added more detail to the methods for the establishment of 2nd generation (page 15, lines 579-585) and statistics to Supplemental Figure S3 using One-Way ANOVA. 2nd generation spheroids showed no statistical differences between treatments. We agree with the reviewer that we need to assess the expression of genes related to self-renewal or stemness. However, as we did not aim to identify a molecular mechanisms in this study and our data only show a correlation with the associated lipid/palmitic acid stimulation, we only focused on functional assays. We plan to investigate molecular changes and the mechanisms in a future study as this was beyond the scope of what we could do during the revision period.

The discussion about Etx in spheroids has been expanded (p 13, lines 456-470) to provide more context for the change in size of spheroids in the presence of FAO inhibition.

  1. Q: Figure S4 is of poor quality, lacks quantification and testing.

R: This figure was only meant to support the findings that 200 mM PA is cytotoxic and inhibits colony formation. As this does not add in any significant way to this study, we deleted this figure from the supplemental material and text.

Minor points

Q: Fig.1a and Fig. 1b please revise significance marks in left panel GC is misplaced under BE and in right panel GC seems significant but unlabeled. PLEASE ANALYZE AND REPORT ALL AGAINST NE CONTROL.

R: Thank you for pointing to the misalignment of the marks in Figure 1. We apologize for not being clear in the legend that the data points highlighted with the asterisk were significant compared to the NE control (only CPT1A in Fig, 1a, b, and CPT1B in Fig. 1a) using a t-test. The composite score compares NE and NA as controls to BE and EAC, but also compares BE and EAC using a t-test, Fig. 1c. We hope with the new alignment those comparisons are better to appreciate.

Q: Fig.2 quantification in (a) misses some samples shown in (b), why?

R: The rationale of the Western Blot was to initially screen all the model cell lines along the Barrett’s tumorigenesis cascade we have in our collection for CPT1A protein expression and move forward with the experiments using select cell lines. We had a prolonged power outage in the building which destroyed some of our storage and left us with only a limited array of cell lines at the time of analyzing the RNA expression. We recently acquired these cell lines again, and given more time, could generate a new Figure 2a after expansion of the cells, isolating RNA and performing new qRT-PCR experiments if required.

Q: Fig.4-d-e-f the Y scale used makes differences hard to appreciate apart for OE33, why not to use log scale?

R: Thank you for the suggestion. We recognize that using the same axis for all the panels makes it difficult to appreciate the differences in response to treatment. We modified the axis by adjusting the maximum value for Fig. 4d and e and used a broken axis to highlight the differences for the EAC cell lines in Fig. 4 f.

Reviewer 2 Report

In the experimental study the rate-limiting enzyme carnitine palmitoyltransferase 1A (CPT1A) is addressed as putative molecular link between lipid-rich hyperalimenation and progression of GERD to Barrett lesions. In the study cell culture, organoids, mice models (IL1beta transgenic and C57BL/6J), and human tissues were investigated. The authors demonstrate data indicating that CPT1A is up regulated in Barrett’s tumorigenesis and that experimental palmitic acid is delivered to mitochondria and promotes cell proliferation and stem cell marker expression.

Comments

1.       There are different types of Barrett mucosa. What type is preferentially addressed with the experiments?

2.       Is there a dynamic increase in the expression of other specialized fatty acid - / lipid transporters by the investigated cells under lipid-rich hyperalimenation?

3.       To my opinion, the molecular mechanism that alimentary provided fatty acids in mitochondria promote cell proliferation and stem cell marker expression is not molecular elucidated. There is description of co-incidence. I recommend to give the conclusions in a more cautious way.

4.       Induction of CPT1A is only found in a subset of cells located in the cardiac junction of squamous and columnar epithelia. The cell type is not clear to the reader. Is there any chance to further characterize the cell type? Is there any evidence for an undifferentiated progenitor cell type?

5.       In the abstract, CPT1A should be introduced to the reader as carnitine palmitoyltransferase 1A.

Author Response

We like to thank the reviewer for the overall positive comments regarding our study “Augmented CPT1A expression is associated with proliferation and colony formation during Barrett’s tumorigenesis.” and the suggestions to improve the manuscript.

Reviewer 1

Major points

  1. Q: Data in Fig.3 are key in the paper but lacks quantification. Quantification and normalization for the cell number is mandatory to evaluate the extent of stain increase upon PA treatment. Moreover, red fluorescence in panel b is hard to appreciate, maybe better to show the two channels separately.

R: The intent of these experiments was to demonstrate that esophageal epithelial cells can take up free fatty acids/lipids such as palmitic acid, which to our knowledge has not been reported before. We state that we can detect the accumulation of lipids with Oil Red O in all cell lines, but detection using the LipidSpotTM shows a more selective increase, especially in BE and EAC cells upon PA stimulation. We therefore focused on the quantification in these experiments and provide a new figure 3 with images for separated channels in panel b and a graph for the quantification, panel c (page 6). Statistics were performed using a t-test with Welch’s correction, and the p-values are indicated in the figure. The materials and methods were expanded to include the quantification (page 17)

  1. Q: Figure 5 lacks statistical hypothesis testing and indications of how many times the experiments were repeated.

R: Thank you for pointing to the lack of statistics. We analyzed all three biological replicates combined for the new data in Figure 5 containing analysis using a One-Way ANOVA. Upon including all replicates, data for IL8 showed an induction upon treatment in STR cells, while the data points for this gene in the other cell lines and for IL1B and CPT1A did not change. We modified the figure (page 9) and the legend to reflect statistical differences as assessed with a One-Way ANOVA.

  1. Q: Figure 6 only shows representative images, blind scoring from an experienced pathologist and statistical testing are required.

R: We agree with the reviewer that we only show representative images and no quantification of statistical testing for the multiplex immunofluorescence stainings. The sections themselves were selected based on review of the tissue pathology for staining yet the 4 color immunofluorescence is difficult to quantify manually. Our assessment with restricted antibody use (as in Figure 1) generates a composite score which combines values for the number of positive cells per section imaged, staining intensity, and co-localization (e.g., same cell or nuclear vs cytoplasmic /membranous). For four colors, this is usually done automatically to generate values ready for statistical analysis, and while we recently acquired the MACSima imaging platform for robustness of quantification, not all antibodies and protocols have been tested for this type of fully automated fluorescence microscopy of multiple samples. We therefore do not make a statement to specific cells being positive for all 4 markers, but rather the association of each within the context of the sample.

  1. Q: Please improve the statistical testing, method description and discussion about ETx a spheroids as this the evidence linking PA and stemness. Expression of stemness related genes in function of FA treatments and ETX exposure is also required.

R: We added more detail to the methods for the establishment of 2nd generation (page 15, lines 579-585) and statistics to Supplemental Figure S3 using One-Way ANOVA. 2nd generation spheroids showed no statistical differences between treatments. We agree with the reviewer that we need to assess the expression of genes related to self-renewal or stemness. However, as we did not aim to identify a molecular mechanisms in this study and our data only show a correlation with the associated lipid/palmitic acid stimulation, we only focused on functional assays. We plan to investigate molecular changes and the mechanisms in a future study as this was beyond the scope of what we could do during the revision period.

The discussion about Etx in spheroids has been expanded (p 13, lines 456-470) to provide more context for the change in size of spheroids in the presence of FAO inhibition.

  1. Q: Figure S4 is of poor quality, lacks quantification and testing.

R: This figure was only meant to support the findings that 200 mM PA is cytotoxic and inhibits colony formation. As this does not add in any significant way to this study, we deleted this figure from the supplemental material and text.

Minor points

Q: Fig.1a and Fig. 1b please revise significance marks in left panel GC is misplaced under BE and in right panel GC seems significant but unlabeled. PLEASE ANALYZE AND REPORT ALL AGAINST NE CONTROL.

R: Thank you for pointing to the misalignment of the marks in Figure 1. We apologize for not being clear in the legend that the data points highlighted with the asterisk were significant compared to the NE control (only CPT1A in Fig, 1a, b, and CPT1B in Fig. 1a) using a t-test. The composite score compares NE and NA as controls to BE and EAC, but also compares BE and EAC using a t-test, Fig. 1c. We hope with the new alignment those comparisons are better to appreciate.

Q: Fig.2 quantification in (a) misses some samples shown in (b), why?

R: The rationale of the Western Blot was to initially screen all the model cell lines along the Barrett’s tumorigenesis cascade we have in our collection for CPT1A protein expression and move forward with the experiments using select cell lines. We had a prolonged power outage in the building which destroyed some of our storage and left us with only a limited array of cell lines at the time of analyzing the RNA expression. We recently acquired these cell lines again, and given more time, could generate a new Figure 2a after expansion of the cells, isolating RNA and performing new qRT-PCR experiments if required.

Q: Fig.4-d-e-f the Y scale used makes differences hard to appreciate apart for OE33, why not to use log scale?

R: Thank you for the suggestion. We recognize that using the same axis for all the panels makes it difficult to appreciate the differences in response to treatment. We modified the axis by adjusting the maximum value for Fig. 4d and e and used a broken axis to highlight the differences for the EAC cell lines in Fig. 4 f.

Round 2

Reviewer 1 Report

Mechanisms are not investigated but improved testing and reporting enhance confidence in the results

Author Response

Thank you for the positive response to our revisions.